# Development of an Electron-Atom Compton Scattering Apparatus Using a Picosecond Pulsed Electron Gun

Yuichi Tachibana, Yuuki Onitsuka, Masakazu Yamazaki [†] and Masahiko Takahashi *

Institute of Multidisciplinary Research for Advanced Materials, Tohoku University, Sendai 980-8577, Japan; yuuichi.tachibana.s7@dc.tohoku.ac.jp (Y.T.); yuuki.onitsuka.e8@tohoku.ac.jp (Y.O.); yamazaki@chem.titech.ac.jp (M.Y.)
* Correspondence: masahiko@tohoku.ac.jp
† Present Address: Department of Chemistry, School of Science, Tokyo Institute of Technology, Tokyo 152-8550, Japan.

**Abstract:** An apparatus has been developed for electron-atom Compton scattering experiments that can employ a pulsed laser and a picosecond pulsed electron beam in a pump-and-probe scheme. The design and technical details of the apparatus are described. Furthermore, experimental results on the Xe atom in its ground state are presented to illustrate the performance of the pulsed electron gun and the detection and spectrometric capabilities for scattered electrons. The scope of future application is also discussed, involving real-time measurement of intramolecular force acting on each constituent atom with different mass numbers, in a transient, evolving system during a molecular reaction.

**Keywords:** electron-atom Compton scattering; time-resolved atomic momentum spectroscopy; intramolecular atomic motion; molecular dynamics

## 1. Introduction

About 20 years have passed since Vos and others [1] suggested that atomic motion in molecules or solids can be directly measured by atomic momentum spectroscopy (AMS), which employs electron-atom Compton scattering at a large scattering angle ($\theta > 90°$) and at incident electron energies of the order of keV or higher. However, experimental AMS data on a gaseous molecule always involve contributions of not only intramolecular atomic motion (vibrational and rotational motions), but also translational motion of the molecule [2–4]. For the purpose of using AMS as a molecular spectroscopic technique, a protocol in data analysis was required, which excludes the contribution of the translational motion from experimental data or extracts the contribution of the intramolecular atomic motion. Such a protocol has been proposed through a series of careful AMS measurements on the influence of translational motion of rare gas atoms [5], and it has successfully been demonstrated for $H_2$ and $D_2$ molecules [6]. As a result, AMS can now be recognized as a new molecular spectroscopic technique, which enables one to measure the intramolecular motion of each constituent atom with different mass numbers. Note that AMS is unique in its ability and it is totally different from existing related techniques, such as laser vibrational spectroscopy that tells one about accurate information regarding frequencies of canonical normal modes of molecular vibration [7], but not regarding atomic motion itself.

The unique ability of AMS would also be useful for studies on molecular dynamics, though such an attempt requires further technical development, namely, time-resolved AMS (TR-AMS). The motivation behind it originates from the fact that canonical normal modes usually include collective motion of many (or all) of the constituent atoms; the displacement of any of the atoms can be observed only in the form of a linear combination of normal modes [8,9], whilst a constituent atom sees only its immediate surroundings as chemistry is local owing to the short-range nature of chemical forces [10]. On the experimental side, however, TR-AMS absolutely requires being equipped with a short-pulsed electron gun. For instance, TR-AMS experiments may seek to employ a laser pulse

and an electron pulse in a pump-and-probe scheme. Nevertheless, it is not an easy task to combine the AMS technique with a short-pulsed electron beam; if the intensity of an electron beam with a short pulse width (<~1 picosecond) is increased, space charge effects may significantly broaden not only the temporal width but also the energy spread of the electron packet [11]. In this respect, however, recent development of a multichannel AMS technique [12] is highly encouraging, because it has successfully improved the instrumental sensitivity as much as possible by almost completely covering the azimuthal angle range available for the electron-atom Compton scattering at a scattering angle $\theta$ of 135°. Recent achievements of time-resolved electron momentum spectroscopy (TR-EMS) [13–15] are also encouraging, which have succeeded in imaging the spatial pattern of the highest occupied molecular orbital of a short-lived molecular excited state using EMS or electron-electron Compton scattering [16,17].

Under the above-mentioned circumstances, the first TR-AMS apparatus has very recently been constructed, which combines the multichannel AMS technique [12] with a picosecond pulsed electron gun. In the present paper, the instrument design and technical details of the apparatus are reported. Furthermore, AMS data for the neutral Xe atom in its ground state (i.e., with no use of the pump laser) have been measured by using the pulsed electron gun and they are presented to illustrate the potential performance of the apparatus for future molecular dynamics studies.

## 2. Theoretical Background

Within the plane-wave impulse approximation (PWIA) [2,18], the AMS scattering process can be described as a billiard-ball-type collision of the incident electron with an atom. Namely, the scattering atom is treated as a "single" free particle, even in the case when it is one of a molecule's constituent atoms, so that it absorbs all of the momentum transfer $K$ (= $p_0 - p_1$) with $p_j$'s ($j = 0, 1$) being the momentum of the incident and quasi-elastically backscattered electrons. The scattering atom with mass $M$ and initial momentum $P$ thus acquires a recoil energy $E_{\text{recoil}}$ through the collision, which is given by

$$E_{\text{recoil}} = \frac{(P + K)^2}{2M} - \frac{P^2}{2M} = \frac{K^2}{2M} + \frac{P \cdot K}{M}. \tag{1}$$

Here the first term on the right-hand side of Equation (1) is the mean recoil energy, $E_{\text{mean}}$, that represents the recoil for scattering from a stationary atom. The second term is the Doppler broadening due to the motion of the scattering atom before collision. Furthermore, since the incident electron must lose the energy equal to $E_{\text{recoil}}$ through the collision to satisfy the law of energy conservation, energy analysis of the scattered electrons can provide direct information about $P$ of the target atom in the form of its projection onto $K$. Such an energy analysis can be made by measuring the electron energy loss $E_{\text{loss}}$ (= $E_0 - E_1$) with $E_j$'s ($j = 0, 1$) being the kinetic energy of the incident and quasi-elastically backscattered electrons, if the $E_0$ value is fixed.

From Equation (1), one can also see the following feature of AMS spectra. For an atomic target, the energy loss spectrum shows only one band centered at an $E_{\text{mean}}$ value associated with its mass $M$. Here the source of the Doppler broadening is only the translational motion of the target atom. On the other hand, for a molecular target, the energy loss spectrum exhibits one or more bands, each centered at $E_{\text{mean}}$ values associated with their own mass. As a result, information about $P$ can be separately obtained for each constituent atom with different mass numbers, if the energy resolution is fine enough to resolve the adjacent bands. In these cases, the Doppler broadening is caused by not only the translational motion of the target molecule but also intramolecular motion of the constituent atom.

## 3. Apparatus

The major components of the TR-AMS apparatus developed here is a femtosecond laser system, a vacuum chamber, a target gas nozzle, a pulsed electron gun, a Faraday cup, a multichannel AMS spectrometer, and a data gathering system, as shown in Figure 1.

Briefly, 5-kHz output from the femtosecond laser is split into a pump path and an electron-generation path. On one hand, the pump laser proceeds, after its repetition rate is halved, into the vacuum chamber so that it interacts with atoms or molecules in the target gas beam. On the other hand, the electron-generation laser is directed toward a back-illuminated photocathode, where the laser generates electron pulses via the photoelectric effect. The resulting electron pulses are then accelerated up to electron energies of the order of keV or higher. They are subsequently used as a pulsed incident electron beam to probe the target atoms or molecules in the laser-gas beam interaction region using the AMS spectrometer with a scattering angle $\theta$ of 135°. The pulsed incident electron beam is collected by the Faraday cup. The time delay between the arrival of the pump laser pulse and the probe electron pulse is controlled with a computer-driven translation stage. The AMS scattering events are recorded with the help of the data gathering electronics system. Details of the individual components of the apparatus are described in Sections 3.1–3.7.

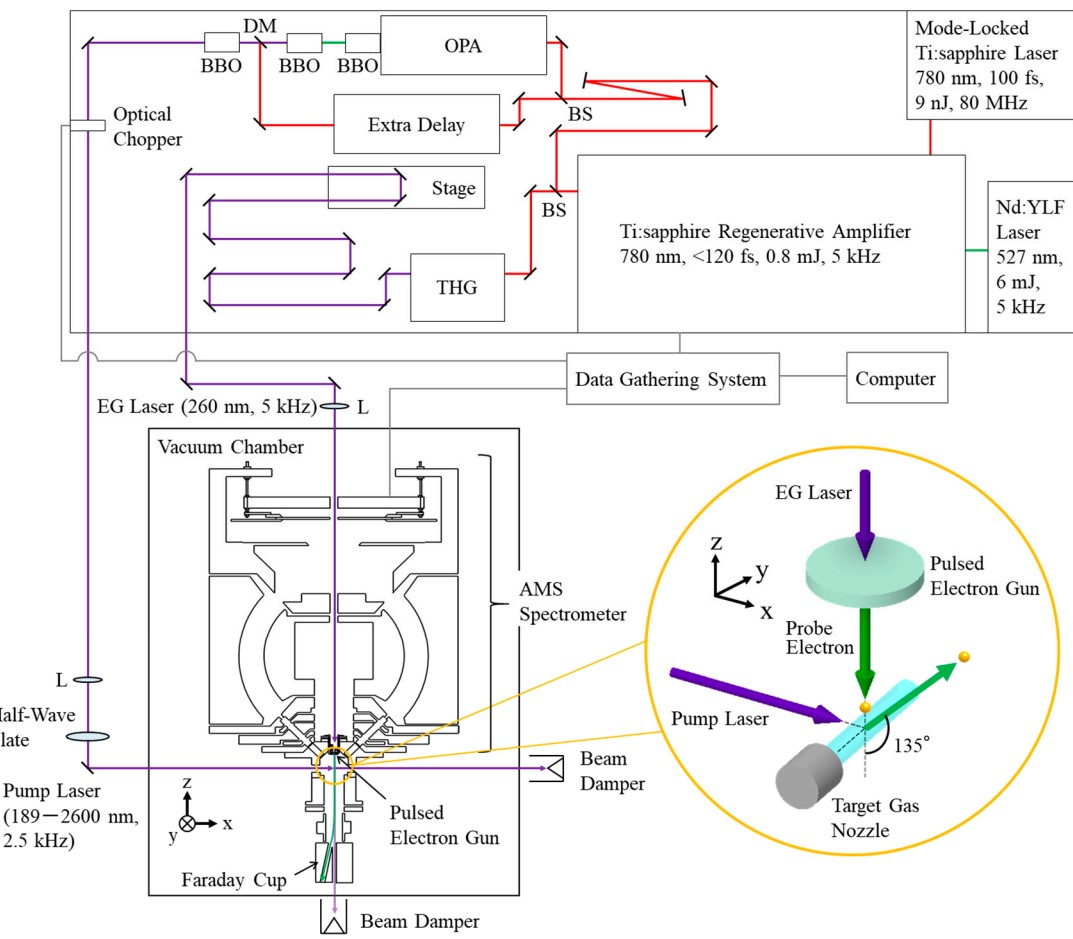

**Figure 1.** Schematic of a time-resolved atomic momentum spectroscopy (TR-AMS), consisting of a femtosecond laser system, a vacuum chamber, a target gas nozzle, a pulsed electron gun, a Faraday cup, a multichannel AMS spectrometer, and a data gathering system. Abbreviations are as follows: BS, beam splitter; OPA, optical parametric amplifier; THG, third harmonic generator; BBO, β-barium borate; DM, dichroic mirror; L, lens; EG, electron-generation.

### 3.1. Femtosecond Laser System

The femtosecond laser system employed is essentially the same as that of the TR-EMS apparatus [13], so only a brief account of it is given here. Femtosecond laser pulses (80 MHz, 9 nJ) centered at 780 nm are generated with a mode-locked Ti: sapphire oscillator (Mai Tai VF, Spectra-Physics). These pulses are then amplified in a 5-kHz Ti: sapphire amplifier (Spitfire Pro XP, Spectra-Physics), which is pumped by a 5-kHz Nd: YLF laser (Empower 30, Spectra-Physics), to produce an output pulse energy of 0.8 mJ. Single-shot autocorrelation of the amplified pulses yields a pulse duration of less than 120 fs. An optical beam splitter

with a splitting ratio of 9:1 is used to divide the 780 nm output into two arms to form the pump laser and the electron-generation laser.

Most (90%) of the 780 nm output is used to produce the pump laser with an optical parametric amplifier (TOPAS-F-DUV, Spectra-Physics), whose wavelength can be tuned over the 189–2600 nm range. Alternatively, the 90% output can be used to create a more intense pump laser (195 nm) with a fourth harmonic generator (TP-FHG-F-HP, Spectra-Physics). The 5-kHz repetition rate of the pump laser is halved by using an optical chopper (Model 3501, New Focus). The polarization of the resulting 2.5-kHz pump laser can be aligned in any direction by rotating a half-wave plate, before the laser interacts with the target atoms and molecules at the scattering point. The remaining small fraction (10%) of the 780 nm output with the 5-kHz repetition rate is frequency tripled in a third harmonic generator (TP-THG-F-HP, Spectra-Physics) to give the electron-generation laser (260 nm, <10 μJ). The propagation time of the electron-generation laser is adjusted in a computer-driven translation stage (FS-3400X, Sigma-Tech) before reaching the photocathode of the pulsed electron gun.

### 3.2. Vacuum Chamber

The vacuum chamber employed is essentially the same as that of the traditional AMS apparatus [12]. It houses a set of components needed for TR-AMS measurements: the target gas nozzle, the pulsed electron gun, the Faraday cup, and the multichannel AMS spectrometer. The chamber is made of nonmagnetic materials such as AISI 310S stainless steel and 2017 aluminum. Its internal dimensions are 412 mm (W) × 665 mm (D) × 400 mm (H). It is evacuated with a molecular turbo pump (3300 $dm^3 s^{-1}$), with an achieved base pressure of $1 \times 10^{-5}$ Pa. The magnetic fields are reduced to lower than 10 mG by a $\mu$-metal shield that covers almost the entire interior of the chamber. $CaF_2$ and synthetic fused silica are used as the chamber window of the pump laser and electron-generation laser, respectively.

### 3.3. Target Gas Nozzle

A continuous flow of sample gas is injected to the scattering point through a single-tube nozzle. The nozzle is made of nonmagnetic, AISI 310S stainless steel with 0.5 mm inner diameter. The nozzle position with respect to the scattering point, where the pulsed incident electron beam of 1 mm diameter intersects perpendicularly with the target gas beam, can be controlled precisely by using a linear-motion drive system. The nozzle tip is set to typically 1.5 mm away from the scattering point to avoid any interaction of the gas nozzle with the electron beam while aiming to have a signal count rate as high as possible.

### 3.4. Pulsed Electron Gun

In the traditional AMS apparatus [12], the incident electron beam of 1 mm diameter is generated by a thermal electron gun that incorporates a tungsten filament with a Wehnelt electrode, followed by a telefocus triple cylinder lens. On the other hand, the pulsed electron gun developed here is essentially similar in shape to that of the TR-EMS apparatus which can provide a picosecond pulsed electron beam [13]. As shown in Figure 2, it consists of a back-illuminated photocathode, a copper electroformed mesh (CU008720, Goodfellow), and a collimating aperture. The photocathode is made of a thin, silver film (40 nm thickness) deposited on one side of a sapphire substrate (10 mm diameter), and it is negatively biased by a variable high-voltage power supply so that the electron gun can produce a pulsed electron beam of an energy of typically 2.0 keV (up to 5 keV). The grounded copper mesh is located 1 mm away from the photocathode surface to accelerate and extract photoelectrons from the cathode. The extracted electrons then pass through the collimating aperture of 1 mm diameter and travel towards the scattering point.

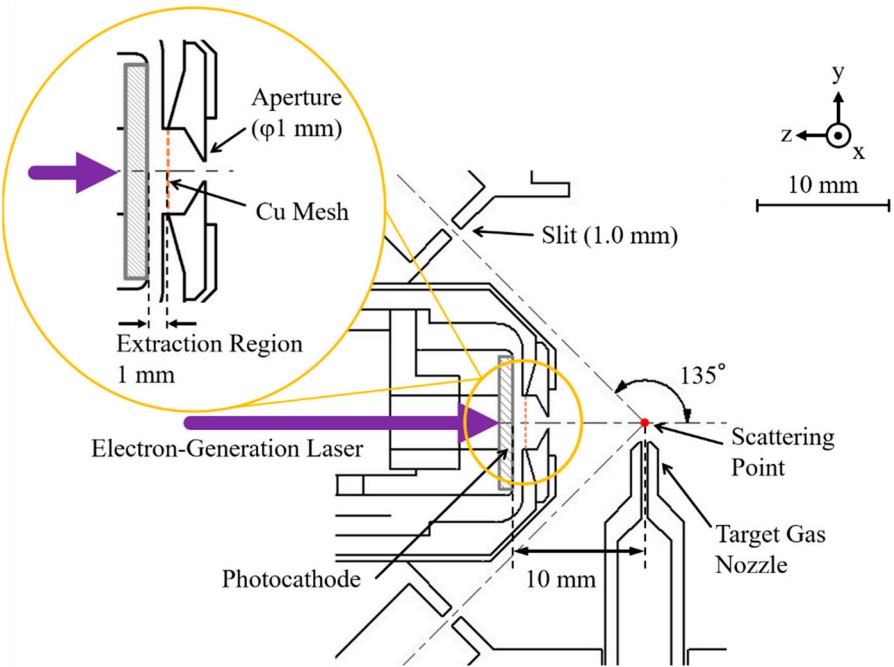

**Figure 2.** Schematic cross section of a pulsed electron gun, consisting of a back-illuminated photo-cathode, a copper electroformed mesh, and a collimating aperture. See text for details.

It should be noted, however, that combining a picosecond pulsed electron gun with an AMS spectrometer is as straightforward as it appears; the difference in the pulsed electron gun between the TR-EMS and TR-AMS apparatuses arises from the difference in their electron scattering geometries. Namely, whilst TR-EMS measures outgoing electrons emerging at a forward scattering angle of 45°, TR-AMS does those at a backward angle of 135°. This difference in geometry raises additional issues to be tackled in the present work. Firstly, since such a picosecond pulsed electron beam becomes significantly broader in kinetic energy distribution as it propagates [11], the pulsed electron gun should be placed as close as possible to the scattering point. In other words, the pulsed electron gun must be small enough so that it can be placed in a limited space inside the first entrance aperture-forming electrode of the AMS spectrometer, as can be seen in Figure 2. Moreover, because of there being no space to insert electrostatic deflectors, the parallelism between the photocathode surface and the copper electroformed mesh upon assembly is all about making sure that the pulsed electron beam propagates to the scattering point properly, though it is more difficult to achieve high precision in the alignment of the electron-gun parts with more miniaturization. Consequently, the sapphire substrate diameter employed here (10 mm) is reduced by 60 percent compared to that of the TR-EMS apparatus (25 mm) and a value of 10 mm has been achieved as the shortest possible distance between the photocathode surface and the scattering point.

*3.5. Faraday Cup*

The intensity of the pulsed electron beam can be measured by a Faraday cup. However, the pulsed electron gun developed here leads to the coaxial propagation of the electron-generation laser and the pulsed electron beam in the same direction. Since the electron-generation laser can produce low-energy secondary electrons from the inside surfaces of the Faraday cup body, it may prevent accurate measurement of the electron beam intensity. Hence the Faraday cup is designed to measure the electron beam intensity only by separating it from the laser light with electrostatic deflection. The laser light is taken out through the chamber window and is damped in the atmosphere.

### 3.6. Multichannel AMS Spectrometer

The multichannel AMS spectrometer developed here is schematically shown in Figure 3. It is similar in principle to our traditional one [12], but is optimized by changing the first entrance aperture width of 0.5 mm and the deceleration ratio of around 20:1 to those of 1.0 mm and 10:1, respectively. These changes are intended to give a workable signal count rate using the pulsed electron gun. Briefly, outgoing electrons emerging from the scattering point are limited by two sets of three entrance apertures in series with 1.0 (first), 0.6 (second), and 2.0 mm widths (third aperture) so that a spherical analyzer accepts those scattered at $\theta = 135° \pm 0.4°$ over azimuthal angle $\phi$ ranges from 0° to 72.5°, from 107.5° to 252.5°, and from 287.5° to 360°. The uncovered $\phi$ ranges are owing to the necessity of the spectrometer pieces to support the inner sphere of the analyzer as well as that of securing the space for the cable routing of the electron gun. Note that in this setup, the entrance slit is formed virtually by the three real entrance apertures and the effective width of the entrance slit is estimated to be ca. 0.6 mm, a value being comparable to that of the traditional spectrometer [12].

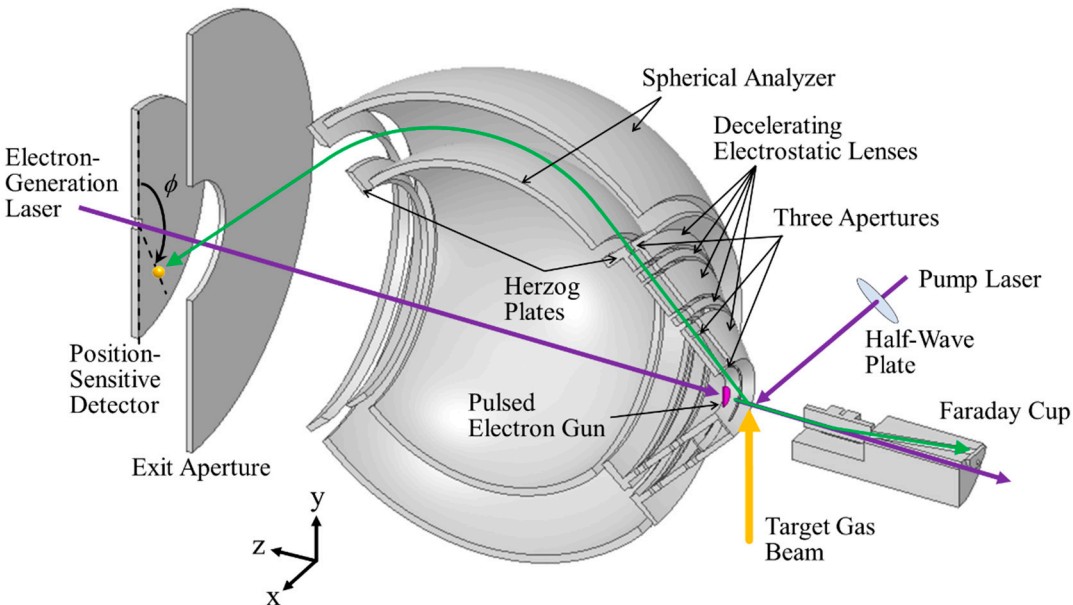

**Figure 3.** Schematic of a multichannel atomic momentum spectroscopy (AMS) spectrometer.

A pair of decelerating electrostatic lenses is used for the scattered electrons to achieve a higher energy resolution. Here the scattered electrons, created at the typical incident electron energy of 2.0 keV, are slowed down to about 200 eV by the decelerating electrostatic lenses consisting of five elements. The initial deceleration stage is formed by the first and second lens elements. The first lens element is at ground potential. The second lens element is operated at typically −1900 V, which significantly decelerates the scattered electrons. The second retarding lens is formed by the third, fourth, and fifth lens elements. These lens elements are operated at potentials of about −1900 V, −1600 V, and −1800 V, respectively. This second retarding lens is designed to also control the focusing point of the spherical analyzer to improve the energy resolution. The fifth lens element also serves the role of the Herzog plate, as discussed below, so the voltage applied to it is determined by the analyzer pass energy of 200 eV.

The design of the spherical analyzer is based on the study by Purcell [19], and the parameters are chosen to be symmetric for ease of construction. The angle of deflection is 90°. The inner and outer radii are 85 and 115 mm, respectively and hence the mean radius is 100 mm. The fringing field compensation is done by using the Herzog plates [20] at both the entrance and exit of the analyzer. The electrons passing through the analyzer are eventually detected with one large-area position-sensitive-detector (PSD) consisting

of micro-channel plates (MCPs) of 120 mm active diameter and three delay-line anodes (HEX120/o, RoentDek) and it is placed behind the exit aperture. This PSD has a central hole to allow for the passing of the electron-generation laser through the detector center [21], as can be seen in Figure 3.

It may be worthwhile to note that the pump laser and pulsed electron beam cross perpendicular to each other. In addition, the polarization of the pump laser can be aligned in any direction in a plane perpendicular to the laser beam direction, as noted earlier. This perpendicular crossing geometry offers an opportunity to study intramolecular atomic motion dynamics of a transient, evolving system, created by the pump laser, as a function of the azimuthal angle $\phi$ and the polarization direction. One illustrative example of this is discussed in Section 5.

### 3.7. Data Gathering System

The data gathering system developed is schematically shown in Figure 4a. For each electron arrival, the HEX120/o detector produces seven separate output pulses, namely, one time-reference pulse from the MCPs and two delayed pulses from the ends of each of the three delay lines for positional readout. These signal pulses are sent through a fast amplifier (FAMP8, RoentDek) and a constant fraction discriminator (CFD8A, RoentDek) to a time-to-digital converter with a time-resolution of 25 ps (TDC8HP, RoentDek) to record their arrival times. The arrival position of each electron on the MCP surface is determined by calculating the arrival time difference between the signal pulses from both ends of each delay line. Trigger signals from the femtosecond laser system and a photo-sensor mounted on the optical chopper head are also recorded together with AMS data in a list mode file. The former signal gives an opportunity to pick out AMS signals or to remove noise signals that are uncorrelated with the timing of laser firing (5 kHz). On the other hand, the latter serves as a tag to identify whether or not the recorded AMS event has occurred with the pump laser (2.5 kHz).

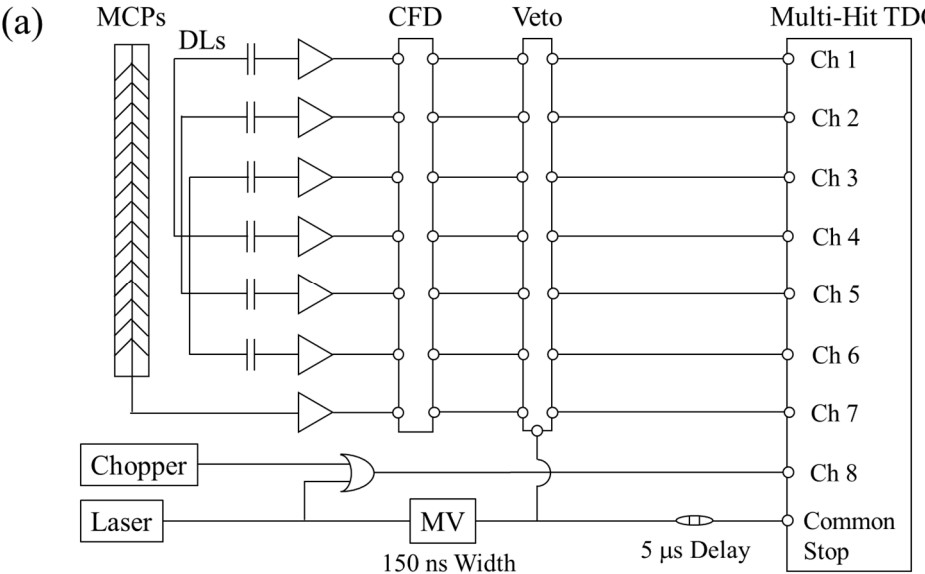

**Figure 4.** *Cont.*

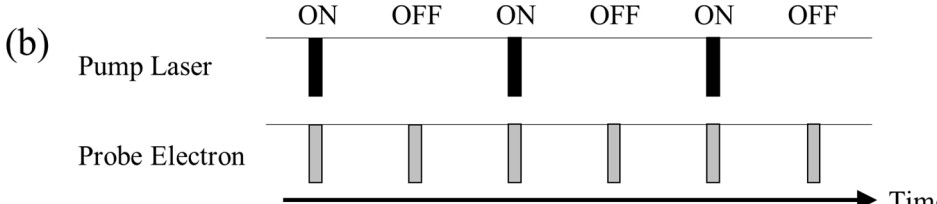

**Figure 4.** (**a**) Schematic diagram of a data gathering system, representing delay lines (DLs), a constant fraction discriminator (CFD), a multivibrator (MV), and a time-to-digital converter (TDC). (**b**) Timing sequence diagram of concurrent measurement of laser-on and laser-off data sets. See text for details.

In the present apparatus, however, it has been found that unexpected noise signals are generated in correlation with the timing of the electron-generation laser shot. Fortunately, the appearance time of such laser noise signals is different from that of the true AMS signals and they are separable from each other, so the laser noise signals are blocked by using the veto function of an octal 100 MHz discriminator (Model 705, Phillips Scientific), as shown in Figure 4a. It should also be noted that since the 5-kHz repetition rate is halved only for the pump laser, the present apparatus concurrently produces two kinds of data sets for future TR-AMS studies on molecular dynamics, as can be seen in Figure 4b. One is data that are measured with the pump laser (laser-on data set). The other is reference data that are measured without the pump laser (laser-off data set) and hence they are equivalent to traditional AMS data for the molecule in its ground state. It is therefore possible to obtain a genuine TR-AMS spectrum by subtracting the laser-off data set with an appropriate weight factor, governed by the laser-excitation probability, from the laser-on data set, while removing contributions of the laser-unexcited molecules. Besides, this concurrent measurement of the laser-on and laser-off data sets is expected to improve the accuracy of the genuine TR-AMS spectrum, as drifts in electron beam current and fluctuations in pump laser intensity as well as target gas density affect it in the same way.

## 4. Performance Testing Results

A series of testing experiments have been carried out to know about the optimized experimental conditions for the TR-AMS apparatus. One such experimental results is presented in Figure 5a, where an image of electrons backscattered quasi-elastically from Xe in its ground state (i.e., with no use of the pump laser) is drawn in a two-dimensional plot. The electron image has been measured by changing the energy of the pulsed electron beam from 1995 to 2004 eV in intervals of 3 eV at a beam intensity of $1 \pm 1$ pA, while the distance between the photocathode surface and the scattering point was set to be 15 mm. The large uncertainty of the electron beam intensity is mainly due to unexpected fluctuations in the background level of a pico-ammeter used, where the phenomenon occurs only when the laser is switched on. In the electron image measurement, the deceleration ratio of 10:1 was employed and the analyzer pass energy was set to collect the 200-eV electrons at the analyzer mean radius of 100 mm. It can be seen from the figure that the electron image covers the azimuthal angle $\phi$ ranges from 0° to 72.5°, from 107.5° to 252.5°, and from 287.5° to 360°, as expected. It can also be seen that a concentric circular image is observed at every incident electron energy value and the circles corresponding to different energies have their own radii. From this relationship between the electron energy and the circle radius, the electron image data can be converted to a one-dimensional energy spectrum, shown in Figure 5b. Four bands corresponding to the incident electron energies employed are evident in the energy spectrum and this observation indicates that AMS experiments using a pulsed electron beam are feasible.

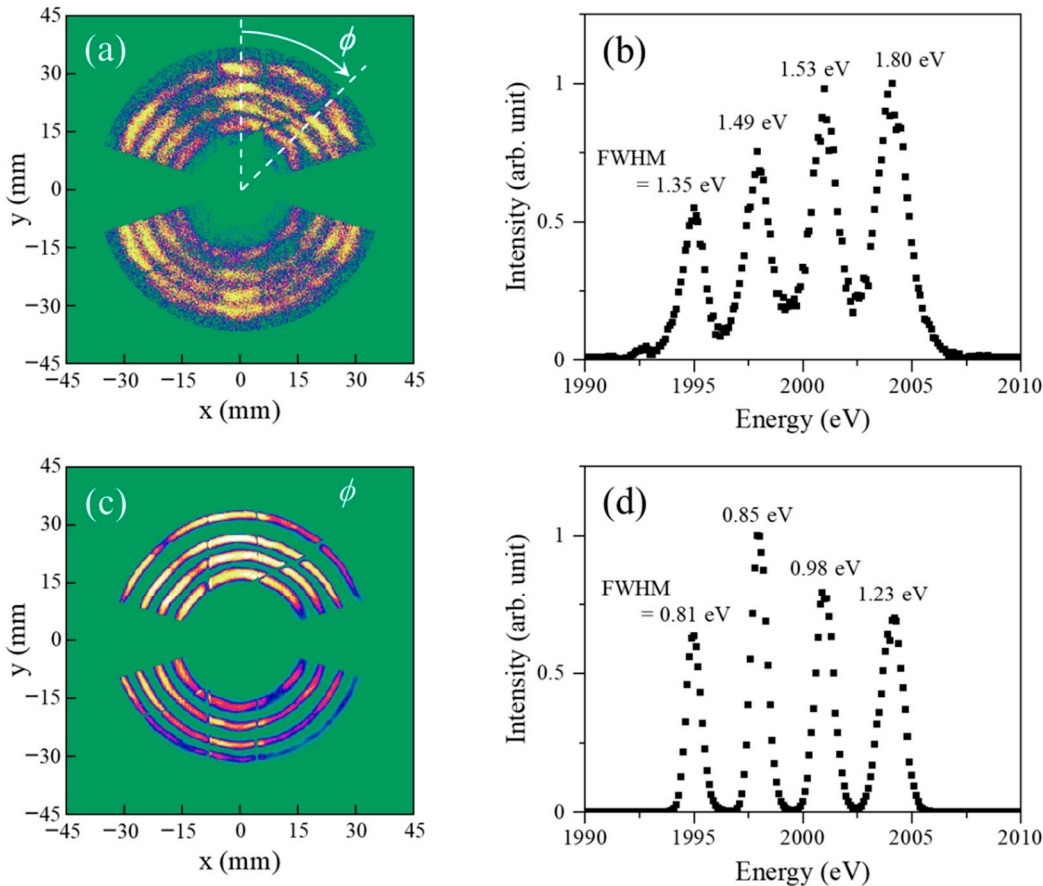

**Figure 5.** (**a**) Two-dimensional image of electrons backscattered quasi-elastically from Xe, measured with the TR-AMS apparatus. (**b**) One-dimensional energy spectrum generated by integration of the two-dimensional data over the azimuthal angle $\phi$. (**c,d**) are those measured with the traditional AMS apparatus [12]. See text for details.

It is also evident from Figure 5b that the full width at the half maximum (FWHM) of the observed bands are found to range between 1.35 and 1.80 eV, depending on the electron energy or electron trajectory path in the spectrometer. The FWHM value of a band in Figure 5b represents the net energy-resolution of the TR-AMS apparatus ($\Delta E_{AMS}$) and it can be analyzed into

$$\Delta E_{AMS} = \sqrt{(\Delta E_0)^2 + (\Delta E_1)^2}, \tag{2}$$

with $\Delta E_0$ and $\Delta E_1$ being the energy spread of the incident electron beam and the energy resolution of the spectrometer, respectively. Furthermore, $\Delta E_1$ can approximately be expressed as

$$\Delta E_1 = (w/2R_0)E_{pass}, \tag{3}$$

where $w$, $R_0$, and $E_{pass}$ are the effective width of the entrance slit (0.6 mm), the analyzer mean radius (100 mm), and pass energy (200 eV), respectively. Hence, $\Delta E_1$ is estimated to be 0.6 eV. Thus, if the 1.49 eV FWHM value of the 1998 eV band is taken as the representative of the four bands, $\Delta E_0$ of the pulsed incident electron beam is calculated to be about 1.4 eV. The value of 1.4 eV is substantially larger than $\Delta E_0$ of a continuous electron beam of the traditional AMS apparatus [5,6,12]. The employed thermal gun is found to provide an incident electron beam with $\Delta E_0$ of ~0.6 eV at intensities of lower than 1 µA. To make this point clear, an electron image and its corresponding one-dimensional energy spectrum are given in Figure 5c,d as a reference, which have been obtained with the same incident electron energies (1995 to 2004 eV in intervals of 3 eV) and the same deceleration ratio (10:1), but with the traditional AMS spectrometer using a continuous incident electron

beam at an intensity of 500 nA. If one does the same analysis, a reasonably consistent value of 0.6 eV is surely obtained as $\Delta E_0$ of the continuous electron beam employed.

In order to access $\Delta E_0$ of the pulsed incident electron beam more closely, consideration of the initial energy spread of the electron packet is also crucial. When a 267 nm laser is used for electron-generation from a silver photocathode, the initial photoelectron energy distribution is often assumed to have a Gaussian transverse profile with 0.6 eV FWHM [11,13]. Thus, the FWHM of the electron packet's initial kinetic energy distribution in the present pulsed electron gun is estimated to be about 0.73 eV, because a 260 nm laser is used here. Clearly, the value of 0.73 eV is substantially smaller than $\Delta E_0$ of 1.4 eV observed in Figure 5b. This argument is unaltered if a different FWHM value of any other band is taken: space charge effects must be responsible for the observation.

Space charge effects can be investigated by simulations using the mean-field model [11], as have been employed for time-resolved electron diffraction [11] and TR-EMS experiments [13]. Figure 6 shows results of such simulations, which illustrates how space charge effects act on an electron packet having a kinetic energy distribution centered at 2.0 keV and travelling in a field free drift region. In the simulations, electron packets were initially prepared as having Gaussian transverse profile containing $N$ = 500, 1250, 2500, 5000, and 10,000 electrons per packet, which correspond to the electron beam intensity of 0.4, 1.0, 2.0, 4.0, and 8.0 pA at the repetition rate of 5 kHz, respectively. Here it was also assumed for the initially prepared electron packets that FWHM of the kinetic energy distribution, the pulse duration, and the beam diameter were 0.73 eV, 120 fs, and 1.0 mm, respectively. Then time evolution of each electron packet was calculated by using the mean-field model. The calculated FWHM of the kinetic energy distribution is plotted against the propagation path length in Figure 6a. In addition, the calculated FWHM of the electron packet pulse duration is shown in Figure 6b.

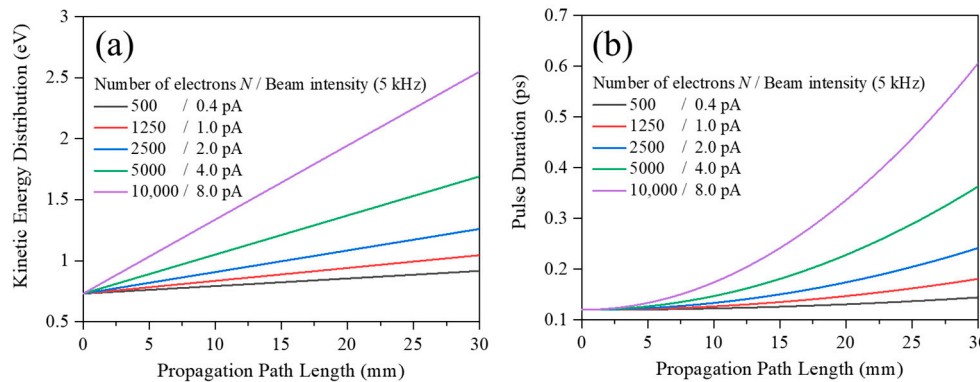

**Figure 6.** (**a**) Full width at the half maximum (FWHM) of the electron packet's kinetic energy distribution (centered at 2.0 keV) and (**b**) FWHM of the electron packet's pulse duration vs. propagation path length and their dependence on the number of electrons, calculated by using the mean field model. See text for details.

It is evident from Figure 6a that the electron packet becomes broader in kinetic energy distribution as it propagates, and this tendency is more prominent with the increase in the number of electrons $N$. If one takes note of the propagation path length of around 15 mm employed in the present experiment, space charge effects are noticeable even at a fairly low beam intensity of 1 pA and the energy spread $\Delta E_0$ reaches the experimentally observed value of 1.4 eV at a beam intensity of several pA. Keeping in mind that the experiment has observed the $\Delta E_0$ value of 1.4 eV at the intensity of $1 \pm 1$ pA, the present simulations are likely to underestimate space charge effects and qualitatively account for the experimental observation. On the other hand, it can be seen from Figure 6b that the temporal width of the pulsed incident electron beam employed in the experiment appears to be smaller than 1 ps at the propagation path length of around 15 mm.

## 5. Discussion

The apparatus developed here has demonstrated its ability to perform AMS experiments using a picosecond pulsed incident electron beam, opening the door to future TR-AMS studies on molecular dynamics or real-time probe of intramolecular atomic motion in a transient, evolving system. There is, however, ample room for improvements, mainly in the pulsed electron beam intensity. The most straightforward way is to make use of the later version of a laser system with a higher repetition rate, which promises to achieve a higher electron beam intensity, while keeping the energy spread $\Delta E_0$ moderate. However, the moderate energy spread of $\Delta E_0$ and the resulting net energy-resolution $\Delta E_{\text{AMS}}$ of about 1.5 eV lead to limitations in the choice of target molecular systems. From this point of view, small molecules containing the lightest atom (H), such as $H_2$, $H_2O$, and hydrocarbons, can be considered to be among the first systems to be studied, because the mean recoil energy $E_{\text{mean}}$ of the H atom is by far the largest and hence the net energy-resolution of about 1.5 eV is still fine enough to resolve the AMS band of a H atom from those of heavier atoms, as is evident from traditional AMS studies on the stable $CH_4$ molecule [2,12]. The idea may be particularly good for H-elimination reaction dynamics, as the kinetic energy or momentum released to the H fragment is usually so large that its AMS band profile could be substantially changed in shape as well as position, compared to that of the H atom constituting the ground-state parent molecule.

In spite of the present limitations mentioned above, however, the TR-AMS apparatus has a lot of scope for its application. In cases of gas phase systems, consider, for example, the very beginning of photo-induced molecular excitation dynamics. Here, Hellmann–Feynman-like electrostatic forces acting on nuclei are generated due to the sudden change of electron cloud upon electronic excitation, and the life of the excited molecule is up to such forces in the very initial stage. Furthermore, because of the anisotropic nature of the molecular photo-absorption cross section, spatial orientation of the excited molecules is not random but is aligned according to $\cos^2\vartheta$ with $\vartheta$ being the angle between the polarization vector of light and the transition moment. Thus, by setting the polarization of the pump laser in a direction, for instance, parallel (perpendicular) to the momentum transfer $K$ vector at the azimuthal angle $\phi$ of 0° (see Figure 3), the present TR-AMS apparatus could probe the parallel (perpendicular) component of such definitive intramolecular forces acting on each constituent atom with different mass numbers in the molecular frame, in a way that the atomic momentum distributions become broader if the force is enhanced upon electronic excitation and vice versa. Furthermore, Ehrenfest's theorem, which relates the time derivative of the expectation value of the momentum operator $p$ to the expectation value of the force, guarantees that TR-AMS would provide a real-time measure of the intramolecular force at any given evolution time.

Not all uses of the TR-AMS apparatus are for gas phase systems: it should be applied to molecular dynamics on surfaces. Since the AMS electron scattering geometry measures outgoing electrons emerging at a backward angle of 135°, there is no limitation in the environment of the target molecule as far as the whole system can be held in a vacuum chamber. As such, the present TR-AMS apparatus and its variants would form a new fundamental basis for promoting not only molecular science but also material science.

**Author Contributions:** Conceptualization and methodology, M.T. and M.Y.; apparatus construction, Y.T. and M.Y.; investigation, Y.T., M.Y., and Y.O.; formal analysis, Y.T. and Y.O.; writing—original draft preparation, Y.T. and Y.O.; writing—review and editing, M.T.; supervision, project administration, and funding acquisition, M.T. All authors have read and agreed to the published version of the manuscript.

**Funding:** This work was partially supported by JSPS KAKENHI, Grant Nos. JP25248002, JP15K13615, JP15H03762, JP17K19095, and JP20225001. It was also supported in part by "Dynamic Alliance for Open Innovation Bridging Human, Environment and Materials" from the Ministry of Education, Culture, Sports, Science and Technology of Japan (MEXT).

**Data Availability Statement:** The data presented in this study are available on request from the corresponding author.

**Acknowledgments:** The authors greatly acknowledge O. Jagutzki and A. Czasch at RoentDek for their advice and help in handling the HEX 120/o detector. They also thank the technical staff of the machine shop at the Institute of Multidisciplinary Research for Advanced Materials (IMRAM), Tohoku University, for their expertise and skills in the development of the present apparatus.

**Conflicts of Interest:** The authors declare no conflict of interest.

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
