# Peer review of "Development of an Electron-Atom Compton Scattering Apparatus Using a Picosecond Pulsed Electron Gun"

_atoms, doi:10.3390/atoms9010019_

Round 1

Reviewer 1 Report

The authors presented an apparatus that they have designed for electron-atom Compton scattering experiments. They described its design and technical details together with some examples of performed test measurements on Xe atom. This is essentially an atomic momentum spectrometer which enables a new molecular spectroscopic technique for determining the intramolecular motion of each constituent atom with different mass numbers. This has been achieved in a pump-and-probe scheme and time resolved experiment with a picosecond pulsed electron beam. In the paper a detail descriptions of all constituent parts of TR-AMS are given: Femtosecond laser system; Vacuum chamber; Target gas nozzle; Pulsed electron gun; Faraday cup; Multichannel AMS spectrometer; and Data gathering system. A comprehensive study of the achieved energy resolution of the pulsed versus continuous electron gun is given in the Section of Performance Testing Results. Also space charge effects have been investigated by performing simulations using the mean-field model. The calculated FWHM of the kinetic energy distribution as well as FWHM of the electron packet pulse duration are plotted against the propagation path length. The simulations show that the electron packet becomes broader in kinetic energy distribution as it propagates and the same holds with the increasing number of electrons.

Reviewer 2 Report

This is a well written paper which describes the development and operation of a new apparatus for extending the application of atomic momentum spectroscopy. The discussion is of high technical merit, very detailed in its presentation, and clear and informative to all readers in the field.  The apparatus that they describe represents a highly complicated technical achievement and I think it makes a substantial contribution to the technical development of the field. It promises to extend the insight available from AMS to include measurements of optically excited molecules which will be an exciting prospect.  I believe that the paper can be published by ATOMS in it present form but the authors may like to consider the following minor points _

  • The title accurately reflects the new apparatus that the authors have described here but when it came to the results I was somewhat surprised that they presented (impressive) data on the ground state of an atomic system rather than demonstrating the full features of the 'pump' nature of the experiment through using a molecular target.  This is not a big issue but given that the authors go to some lengths to describe the optical pumping side of the apparatus, they may choose to state that the present work is just a demonstration of the electron production and detection aspects of the apparatus - which I must say are very impressive.  I presume a demonstration of the pumping side of the experiment will follow soon.
  • I would have appreciated some discussion of the electrometry employed to detect and measure the pulsed electron beam current at the pA level.  I note the measurements state 1±1 pA - where does this large uncertainty arise from - although I appreciate that it it not necessarily significant other than in the modelling of the space charge effects perhaps.

Reviewer 3 Report

The paper is essentially a report on instrumentation development and describes in considerable detail the design and implementation of an experimental system for electron-atom Compton scattering experiments. The facility employs a pulsed laser and a picosecond pulsed electron beam in a pump-and-probe scheme.

A short introduction provides an overview of atomic momentum spectroscopy (AMS) achieved through Compton scattering and applied to measurements of intramolecular atomic motions.

The authors report what they state is the first time-resolved AMS (TR-AMS) system by combining a previously developed multichannel AMS system with a picosecond pulsed electron gun. The latter provides a short pulse of relatively high-energy electrons (~2keV), which are ultimately to be directed at right angles to a pump laser excited molecular target.

The design of the overall system and its constituent parts are well illustrated in the figures provided, together with the data gathering system.

The scattered electrons are slowed down to ~ 200eV to enhance the determined energy resolution. A single laser provides both the pump laser beam and the electro-generation laser beam through a 9:1 beam splitter.

The system has not yet, it seems, been tested in a pumped molecular configuration. Rather, this paper describes some early test performance results obtained with ground state Xenon as the calibrating target. The results, which are compared with those of a previous continuous beam AMS system, show that the overall energy resolution performance of the pulsed electron beam TR-AMS system is significantly limited by space charge effects. Figure 5 nicely demonstrates this comparison between the pulsed and continuous electron beam systems. To better understand the space charge effects, the authors have carried out some computer simulations.

In the final discussion, the authors emphasize that the main result is that the apparatus allows time resolved AMS experiments to be performed with a picosecond pulsed electron beam. They suggest various further potential improvements through, for example, use of a higher repetition laser system and concomitant careful attention to space charge effects in order to maintain reasonable energy resolution. They also briefly describe potential applications of the system.

Overall the paper is well written and explained through clear diagrams. The results should be of considerable scientific interest for those involved with atomic momentum spectroscopies.

The authors should strengthen the paper by outlining the extent to which future applications, for example to molecules, might be compromised by the limits to the achievable energy resolution.
